# Learning and Forecasting Opinion Dynamics in Social Networks

**Abir De**[*]     **Isabel Valera**[†]     **Niloy Ganguly**[*]
**Sourangshu Bhattacharya**[*]     **Manuel Gomez-Rodriguez**[†]
IIT Kharagpur[*]     MPI for Software Systems[†]
{abir.de,niloy,sourangshu}@cse.iitkgp.ernet.in
{ivalera,manuelgr}@mpi-sws.org

## Abstract

Social media and social networking sites have become a global pinboard for exposition and discussion of news, topics, and ideas, where social media users often update their opinions about a particular topic by learning from the opinions shared by their friends. In this context, can we learn a data-driven model of opinion dynamics that is able to accurately forecast users' opinions? In this paper, we introduce SLANT, a probabilistic modeling framework of opinion dynamics, which represents users' opinions over time by means of marked jump diffusion stochastic differential equations, and allows for efficient model simulation and parameter estimation from historical fine grained event data. We then leverage our framework to derive a set of efficient predictive formulas for opinion forecasting and identify conditions under which opinions converge to a steady state. Experiments on data gathered from Twitter show that our model provides a good fit to the data and our formulas achieve more accurate forecasting than alternatives.

## 1 Introduction

Social media and social networking sites are increasingly used by people to express their opinions, give their "hot takes", on the latest breaking news, political issues, sports events, and new products. As a consequence, there has been an increasing interest on leveraging social media and social networking sites to sense and forecast *opinions*, as well as understand *opinion dynamics*. For example, political parties routinely use social media to sense people's opinion about their political discourse[1]; quantitative investment firms measure investor sentiment and trade using social media [18]; and, corporations leverage brand sentiment, estimated from users' posts, likes and shares in social media and social networking sites, to design their marketing campaigns[2]. In this context, multiple methods for sensing opinions, typically based on sentiment analysis [21], have been proposed in recent years. However, methods for accurately forecasting opinions are still scarce [7, 8, 19], despite the extensive literature on theoretical models of opinion dynamics [6, 9].

In this paper, we develop a novel modeling framework of opinion dynamics in social media and social networking sites, SLANT[3], which allows for accurate forecasting of individual users' opinions. The proposed framework is based on two simple intuitive ideas: i) users' opinions are *hidden* until they decide to *share* it with their friends (or neighbors); and, ii) users may update their opinions about a particular topic by learning from the opinions *shared* by their friends. While the latter is one of the main underlying premises used by many well-known theoretical models of opinion dynamics [6, 9, 22], the former has been ignored by models of opinion dynamics, despite its relevance on closely related processes such as information diffusion [12].

More in detail, our proposed model represents users' *latent* opinions as continuous-time stochastic processes driven by a set of marked jump stochastic differential equations (SDEs) [14]. Such construction allows each user's latent opinion to be modulated over time by the opinions asynchronously *expressed* by her neighbors as *sentiment* messages. Here, every time a user expresses an opinion by posting a sentiment message, she *reveals* a noisy estimate of her current latent opinion. Then, we exploit a key property of our model, the Markov property, to develop:

  I. An efficient estimation procedure to find the parameters that maximize the likelihood of a set of (millions of) sentiment messages via convex programming.
 II. A scalable simulation procedure to sample millions of sentiment messages from the proposed model in a matter of minutes.
III. A set of novel predictive formulas for efficient and accurate opinion forecasting, which can also be used to identify conditions under which opinions converge to a steady state of consensus or polarization.

Finally, we experiment on both synthetic and real data gathered from Twitter and show that our model provides a good fit to the data and our predictive formulas achieve more accurate opinion forecasting than several alternatives [7, 8, 9, 15, 26].

**Related work.** There is an extensive line of work on theoretical models of opinion dynamics and opinion formation [3, 6, 9, 15, 17, 26]. However, previous models typically share the following limitations: (i) they do not distinguish between latent opinion and sentiment (or expressed opinion), which is a noisy observation of the opinion (*e.g.*, thumbs up/down, text sentiment); (ii) they consider users' opinions to be updated synchronously in discrete time, however, opinions may be updated asynchronously following complex temporal patterns [12]; (iii) the model parameters are difficult to learn from real fine-grained data and instead are set arbitrarily, as a consequence, they provide inaccurate fine-grained predictions; and, (iv) they focus on analyzing only the steady state of the users' opinions, neglecting the transient behavior of real opinion dynamics which allows for opinion forecasting methods. More recently, there have been some efforts on designing models that overcome some of the above limitations and provide more accurate predictions [7, 8]. However, they do not distinguish between opinion and sentiment and still consider opinions to be updated synchronously in discrete time. Our modeling framework addresses the above limitations and, by doing so, achieves more accurate opinion forecasting than alternatives.

## 2 Proposed model

In this section, we first formulate our model of opinion dynamics, starting from the data it is designed for, and then introduce efficient methods for model parameter estimation and model simulation.

**Opinions data.** Given a directed social network $\mathcal{G} = (\mathcal{V}, \mathcal{E})$, we record each message as $e := (u, m, t)$, where the triplet means that the user $u \in \mathcal{V}$ posted a message with sentiment $m$ at time $t$. Given a collection of messages $\{e_1 = (u_1, m_1, t_1), \ldots, e_n = (u_n, m_n, t_n)\}$, the history $\mathcal{H}_u(t)$ gathers all messages posted by user $u$ up to but not including time $t$, i.e.,

$$\mathcal{H}_u(t) = \{e_i = (u_i, m_i, t_i) | u_i = u \text{ and } t_i < t\}, \tag{1}$$

and $\mathcal{H}(t) := \cup_{u \in \mathcal{V}} \mathcal{H}_u(t)$ denotes the entire history of messages up to but not including time $t$.

**Generative process.** We represent users' latent opinions as a multidimensional stochastic process $\mathbf{x}^*(t)$, in which the $u$-th entry, $x_u^*(t) \in \mathbb{R}$, represents the opinion of user $u$ at time $t$ and the sign $^*$ means that it may depend on the history $\mathcal{H}(t)$. Then, every time a user $u$ posts a message at time $t$, we draw its sentiment $m$ from a sentiment distribution $p(m|x_u^*(t))$. Here, we can also think of the sentiment $m$ of each message as samples from a noisy stochastic process $m_u(t) \sim p(m_u(t)|x_u^*(t))$. Further, we represent the message times by a set of counting processes. In particular, we denote the set of counting processes as a vector $\boldsymbol{N}(t)$, in which the $u$-th entry, $N_u(t) \in \{0\} \cup \mathbb{Z}^+$, counts the number of sentiment messages user $u$ posted up to but not including time $t$. Then, we can characterize the message rate of the users using their corresponding conditional intensities as

$$\mathbb{E}[d\boldsymbol{N}(t) \,|\, \mathcal{H}(t)] = \boldsymbol{\lambda}^*(t)\, dt, \tag{2}$$

where $d\boldsymbol{N}(t) := (\, dN_u(t) \,)_{u \in \mathcal{V}}$ denotes the number of messages per user in the window $[t, t + dt)$ and $\boldsymbol{\lambda}^*(t) := (\, \lambda_u^*(t) \,)_{u \in \mathcal{V}}$ denotes the associated user intensities, which may depend on the history $\mathcal{H}(t)$. We denote the set of user that $u$ *follows* by $\mathcal{N}(u)$. Next, we specify the the intensity functions $\boldsymbol{\lambda}^*(t)$, the dynamics of the users' opinions $\mathbf{x}^*(t)$, and the sentiment distribution $p(m|x_u^*(t))$.

**Intensity for messages.** There is a wide variety of message intensity functions one can choose from to model the users' intensity $\boldsymbol{\lambda}^*(t)$ [1]. In this work, we consider two of the most popular functional forms used in the growing literature on social activity modeling using point processes [10, 24, 5]:

I. **Poisson process.** The intensity is assumed to be independent of the history $\mathcal{H}(t)$ and constant, *i.e.*, $\lambda_u^*(t) = \mu_u$.

II. **Multivariate Hawkes processes.** The intensity captures a mutual excitation phenomena between message events and depends on the whole history of message events $\cup_{v \in \{u \cup \mathcal{N}(u)\}} \mathcal{H}_v(t)$ before $t$:

$$\lambda_u^*(t) = \mu_u + \sum_{v \in u \cup \mathcal{N}(u)} b_{vu} \sum_{e_i \in \mathcal{H}_v(t)} \kappa(t - t_i) = \mu_u + \sum_{v \in u \cup \mathcal{N}(u)} b_{vu} \left( \kappa(t) \star dN_v(t) \right), \quad (3)$$

where the first term, $\mu_u \geqslant 0$, models the publication of messages by user $u$ on her own initiative, and the second term, with $b_{vu} \geqslant 0$, models the publication of additional messages by user $u$ due to the influence that previous messages posted by the users she follows have on her intensity. Here, $\kappa(t) = e^{-\nu t}$ is an exponential triggering kernel modeling the decay of influence of the past events over time and $\star$ denotes the convolution operation.

In both cases, the couple $(\boldsymbol{N}(t), \boldsymbol{\lambda}^*(t))$ is a Markov process, *i.e.*, future states of the process (conditional on past and present states) depends only upon the present state, and we can express the users' intensity more compactly using the following jump stochastic differential equation (SDE):

$$d\boldsymbol{\lambda}^*(t) = \nu(\boldsymbol{\mu} - \boldsymbol{\lambda}^*(t))dt + \mathbf{B}d\boldsymbol{N}(t),$$

where the initial condition is $\boldsymbol{\lambda}^*(0) = \boldsymbol{\mu}$. The Markov property will become important later.

**Stochastic process for opinion.** The opinion $x_u^*(t)$ of a user $u$ at time $t$ adopts the following form:

$$x_u^*(t) = \alpha_u + \sum_{v \in \mathcal{N}(u)} a_{vu} \sum_{e_i \in \mathcal{H}_v(t)} m_i g(t - t_i) = \alpha_u + \sum_{v \in \mathcal{N}(u)} a_{vu} \left( g(t) \star (m_v(t)dN_v(t)) \right), \quad (4)$$

where the first term, $\alpha_u \in \mathbb{R}$, models the original opinion a user $u$ starts with, the second term, with $a_{vu} \in \mathbb{R}$, models updates in user $u$'s opinion due to the influence that previous messages with opinions $m_i$ posted by the users that $u$ follows has on her opinion. Here, $g(t) = e^{-\omega t}$ (where $\omega \geqslant 0$) denotes an exponential triggering kernel, which models the decay of influence over time. The greater the value of $\omega$, the greater the user's tendency to retain her own opinion $\alpha_u$. Under this form, the resulting opinion dynamics are Markovian and can be compactly represented by a set of coupled marked jumped stochastic differential equations (proven in Appendix A):

**Proposition 1** *The tuple* $(\mathbf{x}^*(t), \boldsymbol{\lambda}^*(t), \boldsymbol{N}(t))$ *is a Markov process, whose dynamics are defined by the following marked jumped stochastic differential equations (SDE):*

$$d\mathbf{x}^*(t) = \omega(\boldsymbol{\alpha} - \mathbf{x}^*(t))dt + \mathbf{A}(\mathbf{m}(t) \odot d\boldsymbol{N}(t)) \qquad (5)$$

$$d\boldsymbol{\lambda}^*(t) = \nu(\boldsymbol{\mu} - \boldsymbol{\lambda}^*(t))dt + \mathbf{B}\,d\boldsymbol{N}(t) \qquad (6)$$

*where the initial conditions are* $\boldsymbol{\lambda}^*(0) = \boldsymbol{\mu}$ *and* $\mathbf{x}^*(0) = \boldsymbol{\alpha}$*, the marks are the sentiment messages* $\mathbf{m}(t) = \left( m_u(t) \right)_{u \in \mathcal{V}}$*, with* $m_u(t) \sim p(m|x_u^*(t))$*, and the sign* $\odot$ *denotes pointwise product.*

The above mentioned Markov property will be the key to the design of efficient model parameter estimation and model simulation algorithms.

**Sentiment distribution.** The particular choice of sentiment distribution $p(m|x_u^*(t))$ depends on the recorded marks. For example, one may consider:

I. **Gaussian Distribution** The sentiment is assumed to be a real random variable $m \in \mathbb{R}$, *i.e.*, $p(m|x_u(t)) = \mathcal{N}(x_u(t), \sigma_u)$. This fits well scenarios in which sentiment is extracted from text using sentiment analysis [13].

II. **Logistic.** The sentiment is assumed to be a binary random variable $m \in \{-1, 1\}$, *i.e.*, $p(m|x_u(t)) = 1/(1 + \exp(-m \cdot x_u(t)))$. This fits well scenarios in which sentiment is measured by means of up votes, down votes or likes.

Our model estimation method can be easily adapted to any log-concave sentiment distribution. However, in the remainder of the paper, we consider the Gaussian distribution since, in our experiments, sentiment is extracted from text using sentiment analysis.

## 2.1 Model parameter estimation

Given a collection of messages $\mathcal{H}(T) = \{(u_i, m_i, t_i)\}$ recorded during a time period $[0, T)$ in a social network $\mathcal{G} = (\mathcal{V}, \mathcal{E})$, we can find the optimal parameters $\boldsymbol{\alpha}$, $\boldsymbol{\mu}$, $\boldsymbol{A}$ and $\boldsymbol{B}$ by solving a maximum likelihood estimation (MLE) problem[4]. To do so, it is easy to show that the log-likelihood of the messages is given by

$$\mathcal{L}(\boldsymbol{\alpha}, \boldsymbol{\mu}, \boldsymbol{A}, \boldsymbol{B}) = \underbrace{\sum_{e_i \in \mathcal{H}(T)} \log p(m_i | x_{u_i}^*(t_i))}_{\text{message sentiments}} + \underbrace{\sum_{e_i \in \mathcal{H}(T)} \log \lambda_{u_i}^*(t_i) - \sum_{u \in \mathcal{V}} \int_0^T \lambda_u^*(\tau) \, d\tau}_{\text{message times}}. \quad (7)$$

Then, we can find the optimal parameters $(\boldsymbol{\alpha}, \boldsymbol{\mu}, \boldsymbol{A}, \boldsymbol{B})$ using MLE as

$$\underset{\boldsymbol{\alpha}, \boldsymbol{\mu} \geq 0, \boldsymbol{A}, \boldsymbol{B} \geq 0}{\text{maximize}} \quad \mathcal{L}(\boldsymbol{\alpha}, \boldsymbol{\mu}, \boldsymbol{A}, \boldsymbol{B}). \quad (8)$$

Note that, as long as the sentiment distributions are log-concave, the MLE problem above is concave and thus can be solved efficiently. Moreover, the problem decomposes in $2|\mathcal{V}|$ independent subproblems, two per user $u$, since the first term in Eq. 7 only depends on $(\boldsymbol{\alpha}, \boldsymbol{A})$ whereas the last two terms only depend on $(\boldsymbol{\mu}, \boldsymbol{B})$, and thus can be readily parallelized. Then, we find $(\boldsymbol{\mu}^*, \boldsymbol{B}^*)$ using spectral projected gradient descent [4], which works well in practice and achieves $\varepsilon$ accuracy in $O(\log(1/\varepsilon))$ iterations, and find $(\boldsymbol{\alpha}^*, \boldsymbol{A}^*)$ analytically, since, for Gaussian sentiment distributions, the problem reduces to a least-square problem. Fortunately, in each subproblem, we can use the Markov property from Proposition 1 to precompute the sums and integrals in (8) in linear time, *i.e.*, $O(|\mathcal{H}_u(T)| + |\cup_{v \in \mathcal{N}(u)} \mathcal{H}_v(T)|)$. Appendix H summarizes the overall estimation algorithm.

## 2.2 Model simulation

We leverage the efficient sampling algorithm for multivariate Hawkes introduced by Farajtabar et al. [11] to design a scalable algorithm to sample opinions from our model. The two key ideas that allow us to adapt the procedure by Farajtabar et al. to our model of opinion dynamics, while keeping its efficiency, are as follows: (i) the opinion dynamics, defined by Eqs. 5 and 6, are Markovian and thus we can update individual intensities and opinions in $O(1)$ – let $t_i$ and $t_{i+1}$ be two consecutive events, then, we can compute $\lambda^*(t_{i+1})$ as $(\lambda^*(t_i) - \mu) \exp(-\nu(t_{i+1} - t_i)) + \mu$ and $x^*(t_{i+1})$ as $(x^*(t_i) - \alpha) \exp(-\omega(t_{i+1} - t_i)) + \alpha$, respectively; and, (ii) social networks are typically sparse and thus both $\boldsymbol{A}$ and $\boldsymbol{B}$ are also sparse, then, whenever a node expresses its opinion, only a small number of opinions and intensity functions in its local neighborhood will change. As a consequence, we can reuse the majority of samples from the intensity functions and sentiment distributions for the next new sample. Appendix I summarizes the overall simulation algorithm.

# 3 Opinion forecasting

Our goal here is developing efficient methods that leverage our model to forecast a user $u$'s opinion $x_u(t)$ at time $t$ given the history $\mathcal{H}(t_0)$ up to time $t_0 < t$. In the context of our probabilistic model, we will forecast this opinion by efficiently computing the conditional expectation $\mathbb{E}_{\mathcal{H}(t) \setminus \mathcal{H}(t_0)}[x_u^*(t) | \mathcal{H}(t_0)]$, where $\mathcal{H}(t) \setminus \mathcal{H}(t_0)$ denotes the average across histories from $t_0$ to $t$, while conditioning on the history up to $\mathcal{H}(t_0)$.

To this aim, we will develop analytical and sampling based methods to compute the above conditional expectation. Moreover, we will use the former to identify under which conditions users' average opinion converges to a steady state and, if so, find the steady state opinion. In this section, we write $\mathcal{H}_t = \mathcal{H}(t)$ to lighten the notation and denote the eigenvalues of a matrix $\boldsymbol{X}$ by $\xi(\boldsymbol{X})$.

## 3.1 Analytical forecasting

In this section, we derive a set of formulas to compute the conditional expectation for both Poisson and Hawkes messages intensities. However, since the derivation of such formulas for general multivariate Hawkes is difficult, we focus here on the case when $b_{vu} = 0$ for all $v, u \in \mathcal{G}, v \neq u$, and rely on the efficient sampling based method for the general case.

**I. Poisson intensity.** Consider each user's messages follow a Poisson process with rate $\mu_u$. Then, the conditional average opinion is given by (proven in Appendix C):

**Theorem 2** *Given a collection of messages $\mathcal{H}_{t_0}$ recorded during a time period $[0, t_0)$ and $\lambda_u^*(t) = \mu_u$ for all $u \in \mathcal{G}$, then,*

$$\mathbb{E}_{\mathcal{H}_t \backslash \mathcal{H}_{t_0}}[\boldsymbol{x}^*(t)|\mathcal{H}_{t_0}] = e^{(\boldsymbol{A}\boldsymbol{\Lambda}_1 - \omega I)(t-t_0)}\boldsymbol{x}(t_0) + \omega(\boldsymbol{A}\boldsymbol{\Lambda}_1 - \omega I)^{-1}\left(e^{(\boldsymbol{A}\boldsymbol{\Lambda}_1 - \omega I)(t-t_0)} - \boldsymbol{I}\right)\boldsymbol{\alpha}, \tag{9}$$

*where $\boldsymbol{\Lambda}_1 := \mathrm{diag}[\boldsymbol{\mu}]$ and $(\boldsymbol{x}(t_0))_{u \in \mathcal{V}} = \alpha_u + \sum_{v \in \mathcal{N}(u)} a_{uv} \sum_{t_i \in \mathcal{H}_v(t_0)} e^{-\omega(t_0 - t_i)} m_v(t_i)$.*

Remarkably, we can efficiently compute both terms in Eq. 9 by using the iterative algorithm by Al-Mohy et al. [2] for the matrix exponentials and the well-known GMRES method [23] for the matrix inversion. Given this predictive formula, we can easily study the stability condition and, for stable systems, find the steady state conditional average opinion (proven in Appendix D):

**Theorem 3** *Given the conditions of Theorem 2, if $Re[\xi(\boldsymbol{A}\boldsymbol{\Lambda}_1)] < \omega$, then,*

$$\lim_{t \to \infty} \mathbb{E}_{\mathcal{H}_t \backslash \mathcal{H}_{t_0}}[\boldsymbol{x}^*(t)|\mathcal{H}_{t_0}] = \left(I - \frac{\boldsymbol{A}\boldsymbol{\Lambda}_1}{w}\right)^{-1}\boldsymbol{\alpha}. \tag{10}$$

The above results indicate that the conditional average opinions are nonlinearly related to the parameter matrix $\boldsymbol{A}$, which depends on the network structure, and the message rates $\boldsymbol{\mu}$, which in this case are assumed to be constant and independent on the network structure. Figure 1 provides empirical evidence of these results.

**II. Multivariate Hawkes Process.** Consider each user's messages follow a multivariate Hawkes process, given by Eq. 3, and $b_{vu} = 0$ for all $v, u \in \mathcal{G}, v \neq u$. Then, the conditional average opinion is given by (proven in Appendix E):

**Theorem 4** *Given a collection of messages $\mathcal{H}_{t_0}$ recorded during a time period $[0, t_0)$ and $\lambda_u^*(t) = \mu_u + b_{uu} \sum_{e_i \in \mathcal{H}_u(t)} e^{-\nu(t-t_i)}$ for all $u \in \mathcal{G}$, then, the conditional average satisfies the following differential equation:*

$$\frac{d\mathbb{E}_{\mathcal{H}_t \backslash \mathcal{H}_{t_0}}[\boldsymbol{x}^*(t)|\mathcal{H}_{t_0}]}{dt} = [-\omega I + \boldsymbol{A}\boldsymbol{\Lambda}(t)]\mathbb{E}_{\mathcal{H}_t \backslash \mathcal{H}_{t_0}}[\boldsymbol{x}^*(t)|\mathcal{H}_{t_0}] + \omega\boldsymbol{\alpha}, \tag{11}$$

*where*

$$\boldsymbol{\Lambda}(t) = \mathrm{diag}\left(\mathbb{E}_{\mathcal{H}_t \backslash \mathcal{H}_{t_0}}[\boldsymbol{\lambda}^*(t)|\mathcal{H}_{t_0}]\right),$$

$$\mathbb{E}_{\mathcal{H}_t \backslash \mathcal{H}_{t_0}}[\boldsymbol{\lambda}^*(t)|\mathcal{H}_{t_0}] = e^{(\boldsymbol{B}-\nu I)(t-t_0)}\boldsymbol{\eta}(t_0) + \nu(\boldsymbol{B}-\nu I)^{-1}\left(e^{(\boldsymbol{B}-\nu\boldsymbol{I})(t-t_0)} - \boldsymbol{I}\right)\boldsymbol{\mu} \quad \forall t \geq t_0,$$

$$(\boldsymbol{\eta}(t_0))_{u \in \mathcal{V}} = \mu_u + \sum_{v \in \mathcal{N}(u)} b_{uv} \sum_{t_i \in \mathcal{H}_v(t_0)} e^{-\nu(t_0 - t_i)},$$

$$\boldsymbol{B} = \mathrm{diag}\left([b_{11}, \ldots, b_{|\mathcal{V}||\mathcal{V}|}]^\top\right).$$

Here, we can compute the conditional average by solving numerically the differential equation above, which is not stochastic, where we can efficiently compute the vector $\mathbb{E}_{\mathcal{H}_t}[\boldsymbol{\lambda}^*(t)]$ by using again the algorithm by Al-Mohy et al. [2] and the GMRES method [23].

In this case, the stability condition and the steady state conditional average opinion are given by (proven in Appendix F):

**Theorem 5** *Given the conditions of Theorem 4, if the transition matrix $\Phi(t)$ associated to the time-varying linear system described by Eq. 11 satisfies that $||\Phi(t)|| \leq \gamma e^{-ct} \ \forall t > 0$, where $\gamma, c > 0$, then,*

$$\lim_{t \to \infty} \mathbb{E}_{\mathcal{H}_t \backslash \mathcal{H}_{t_0}}[\boldsymbol{x}^*(t)|\mathcal{H}_{t_0}] = \left(I - \frac{\boldsymbol{A}\boldsymbol{\Lambda}_2}{w}\right)^{-1}\boldsymbol{\alpha}, \tag{12}$$

*where $\boldsymbol{\Lambda}_2 := \mathrm{diag}\left[\left(I - \frac{\boldsymbol{B}}{\nu}\right)^{-1}\boldsymbol{\mu}\right]$*

The above results indicate that the conditional average opinions are nonlinearly related to the parameter matrices $\boldsymbol{A}$ and $\boldsymbol{B}$. This suggests that the effect of the temporal influence on the opinion evolution, by means of the parameter matrix $\boldsymbol{B}$ of the multivariate Hawkes process, is non trivial. We illustrate this result empirically in Figure 1.

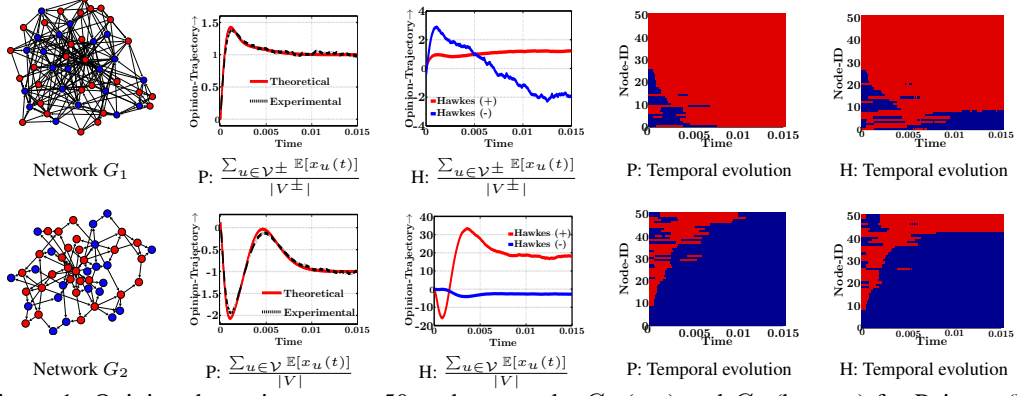

Figure 1: Opinion dynamics on two 50-node networks $G_1$ (top) and $G_2$ (bottom) for Poisson (P) and Hawkes (H) message intensities. The first column visualizes the two networks and opinion of each node at $t = 0$ (positive/negative opinions in red/blue). The second column shows the temporal evolution of the theoretical and empirical average opinion for Poisson intensities. The third column shows the temporal evolution of the empirical average opinion for Hawkes intensities, where we compute the average separately for positive (+) and negative (−) opinions in the steady state. The fourth and fifth columns shows the polarity of average opinion per user over time.

## 3.2 Simulation based forecasting

Given the efficient simulation procedure described in Section 2.2, we can readily derive a general simulation based formula for opinion forecasting:

$$\mathbb{E}_{\mathcal{H}_t \setminus \mathcal{H}_{t_0}}[\boldsymbol{x}^*(t)|\mathcal{H}_{t_0}] \approx \hat{\boldsymbol{x}}^*(t) = \frac{1}{n}\sum_{l=1}^{n}\boldsymbol{x}_l^*(t), \tag{13}$$

where $n$ is the number of times that we simulate the opinion dynamics and $\boldsymbol{x}_l^*(t)$ gathers the users' opinion at time $t$ for the $l$-th simulation. Moreover, we have the following theoretical guarantee (proven in Appendix G):

**Theorem 6** *Simulate the opinion dynamics up to time $t > t_0$ the following number of times:*

$$n \geq \frac{1}{3\epsilon^2}(6\sigma_{max}^2 + 4x_{max}\epsilon)\log(2/\delta), \tag{14}$$

*where $\sigma_{max}^2 = \max_{u \in \mathcal{G}} \sigma_{\mathcal{H}_t \setminus \mathcal{H}_{t_0}}^2(x_u^*(t)|\mathcal{H}_{t_0})$ is the maximum variance of the users' opinions, which we analyze in Appendix G, and $x_{max} \geq |x_u(t)|, \forall u \in \mathcal{G}$ is an upper bound on the users' (absolute) opinions. Then, for each user $u \in \mathcal{G}$, the error between her true and estimated average opinion satisfies that $|\hat{x}_u^*(t) - \mathbb{E}_{\mathcal{H}_t \setminus \mathcal{H}_{t_0}}[x_u^*(t)|\mathcal{H}_{t_0}]| \leq \epsilon$ with probability at least $1 - \delta$.*

## 4 Experiments

### 4.1 Experiments on synthetic data

We first provide empirical evidence that our model is able to produce different types of opinion dynamics, which may or may not converge to a steady state of consensus or polarization. Then, we show that our model estimation and simulation algorithms as well as our predictive formulas scale to networks with millions of users and events. Appendix J contains an evaluation of the accuracy of our model parameter estimation method.

**Different types of opinion dynamics.** We first simulate our model on two different small networks using Poisson intensities, *i.e.*, $\lambda_u^*(t) = \mu_u$, $\mu_u \sim U(0,1)$ $\forall u$, and then simulate our model on the same networks while using Hawkes intensities with $b_{vu} \sim U(0,1)$ on 5% of the nodes, chosen at random, and the original Poisson intensities on the remaining nodes. Figure 1 summarizes the results, which show that (i) our model is able to produce opinion dynamics that converge to consensus (second column) and polarization (third column); (ii) the opinion forecasting formulas described in Section 3 closely match an simulation based estimation (second column); and, (iii) the evolution of

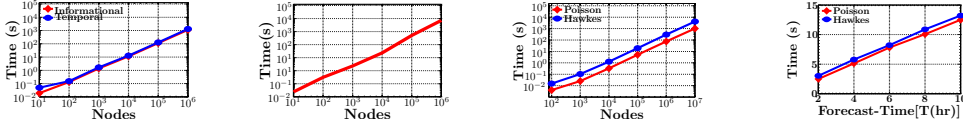

(a) Estimation vs # nodes    (b) Simulation vs # nodes    (c) Forecast vs # nodes    (d) Forecast vs $T$

Figure 2: Panels (a) and (b) show running time of our estimation and simulation procedures against number of nodes, where the average number of events per node is 10. Panels (c) and (d) show the running time needed to compute our analytical formulas against number of nodes and time horizon $T = t - t_0$, where the number of nodes is $10^3$. In Panel (c), $T = 6$ hours. For all panels, the average degree per node is 30. The experiments are carried out in a single machine with 24 cores and 64 GB of main memory.

the average opinion and whether opinions converge to a steady state of consensus or polarization depend on the functional form of message intensity[5].

**Scalability.** Figure 2 shows that our model estimation and simulation algorithms, described in Sections 2.1 and 2.2, and our analytical predictive formulas, described in Section 3.1, scale to networks with millions of users and events. For example, our algorithm takes 20 minutes to estimate the model parameters from 10 million events generated by one million nodes using a single machine with 24 cores and 64 GB RAM.

### 4.2 Experiments on real data

We use real data gathered from Twitter to show that our model can forecast users' opinions more accurately than six state of the art methods [7, 8, 9, 15, 19, 26] (see Appendix L).

**Experimental Setup.** We experimented with five Twitter datasets about current real-world events (Politics, Movie, Fight, Bollywood and US), in which, for each recorded message $i$, we compute its sentiment value $m_i$ using a popular sentiment analysis toolbox, specially designed for Twitter [13]. Here, the sentiment takes values $m \in (-1, 1)$ and we consider the sentiment polarity to be simply $\text{sign}(m)$. Appendix K contains further details and statistics about these datasets.

**Opinion forecasting.** We first evaluate the performance of our model at predicting sentiment (expressed opinion) at a message level. To do so, for each dataset, we first estimate the parameters of our model, SLANT, using messages from a training set containing the (chronologically) first 90% of the messages. Here, we set the decay parameters of the exponential triggering kernels $\kappa(t)$ and $g(t)$ by cross-validation. Then, we evaluate the predictive performance of our opinion forecasting formulas using the last 10% of the messages[6]. More specifically, we predict the sentiment value $m$ for each message posted by user $u$ in the test set given the history up to $T$ hours before the time of the message as $\hat{m} = E_{\mathcal{H}_t \setminus \mathcal{H}_{t-T}}[x_u^*(t)|\mathcal{H}_{t-T}]$. We compare the performance of our model with the asynchronous linear model (AsLM) [8], DeGroot's model [9], the voter model [26], the biased voter model [7], the flocking model [15], and the sentiment prediction method based on collaborative filtering by Kim et al. [19], in terms of: (i) the mean squared error between the true ($m$) and the estimated ($\hat{m}$) sentiment value for all messages in the held-out set, *i.e.*, $\mathbb{E}[(m - \hat{m})^2]$, and (ii) the failure rate, defined as the probability that the true and the estimated polarity do not coincide, *i.e.*, $\mathbb{P}(\text{sign}(m) \neq \text{sign}(\hat{m}))$. For the baselines algorithms, which work in discrete time, we simulate $N_T$ rounds in $(t - T, t)$, where $N_T$ is the number of posts in time $T$. Figure 3 summarizes the results, which show that: (i) our opinion forecasting formulas consistently outperform others both in terms of MSE (often by an order of magnitude) and failure rate;[7] (ii) its forecasting performance degrades gracefully with respect to $T$, in contrast, competing methods often fail catastrophically; and, (iii) it achieves an additional mileage by using Hawkes processes instead of Poisson processes. To some extent, we believe SLANT's superior performance is due to its ability to leverage historical data to learn its model parameters and then simulate realistic temporal patterns.

Finally, we look at the forecasting results at a network level and show that our forecasting formulas can also predict the evolution of opinions macroscopically (in terms of the average opinion across users). Figure 4 summarizes the results for two real world datasets, which show that the forecasted

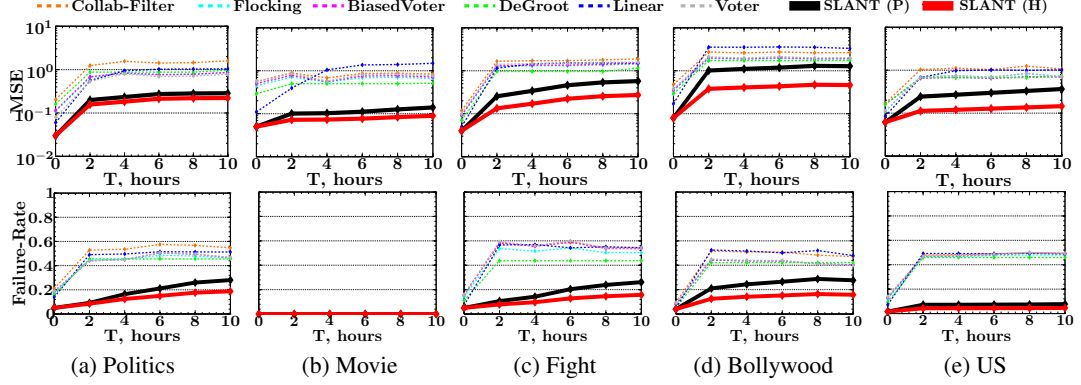

(a) Politics      (b) Movie      (c) Fight      (d) Bollywood      (e) US

Figure 3: Sentiment prediction performance using a 10% held-out set for each real-world dataset. Performance is measured in terms of mean squared error (MSE) on the sentiment value, $\mathbb{E}[(m - \hat{m})^2]$, and failure rate on the sentiment polarity, $\mathbb{P}(\text{sign}(m) \neq \text{sign}(\hat{m}))$. For each message in the held-out set, we predict the sentiment value $m$ given the history up to $T$ hours before the time of the message, for different values of $T$. Nowcasting corresponds to $T = 0$ and forecasting to $T > 0$. The sentiment value $m \in (-1, 1)$ and the sentiment polarity sign $(m) \in \{-1, 1\}$.

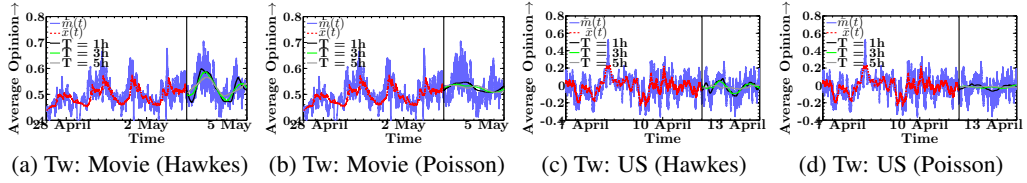

(a) Tw: Movie (Hawkes)   (b) Tw: Movie (Poisson)   (c) Tw: US (Hawkes)   (d) Tw: US (Poisson)

Figure 4: Macroscopic sentiment prediction given by our model for two real-world datasets. The panels show the observed sentiment $\bar{m}(t)$ (in blue, running average), inferred opinion $\bar{x}(t)$ on the training set (in red), and forecasted opinion $\mathbb{E}_{\mathcal{H}_t \setminus \mathcal{H}_{t-T}}[x_u(t)|\mathcal{H}_{t-T}]$ for $T = 1, 3$, and 5 hours on the test set (in black, green and gray, respectively), where the symbol $^-$ denotes average across users.

opinions become less accurate as the time $T$ becomes larger, since the average is computed on longer time periods. As expected, our model is more accurate when the message intensities are modeled using multivariate Hawkes. We found qualitatively similar results for the remaining datasets.

## 5   Conclusions

We proposed a modeling framework of opinion dynamics, whose key innovation is modeling users' *latent* opinions as continuous-time stochastic processes driven by a set of marked jump stochastic differential equations (SDEs) [14]. Such construction allows each user's latent opinion to be modulated over time by the opinions asynchronously *expressed* by her neighbors as *sentiment* messages. We then exploited a key property of our model, the Markov property, to design efficient parameter estimation and simulation algorithms, which scale to networks with millions of nodes. Moreover, we derived a set of novel predictive formulas for efficient and accurate opinion forecasting and identified conditions under which opinions converge to a steady state of consensus or polarization. Finally, we experimented with real data gathered from Twitter and showed that our framework achieves more accurate opinion forecasting than state-of-the-arts.

Our model opens up many interesting venues for future work. For example, in Eq. 4, our model assumes a linear dependence between users' opinions, however, in some scenarios, this may be a coarse approximation. A natural follow-up to improve the opinion forecasting accuracy would be considering nonlinear dependences between opinions. It would be interesting to augment our model to jointly consider correlations between different topics. One could leverage our modeling framework to design opinion shaping algorithms based on stochastic optimal control [14, 25]. Finally, one of the key modeling ideas is realizing that users' expressed opinions (be it in the form of thumbs up/down or text sentiment) can be viewed as noisy discrete samples of the users' latent opinion localized in time. It would be very interesting to generalize this idea to any type of event data and derive sampling theorems and conditions under which an underlying general continuous signal of interest (be it user's opinion or expertise) can be recovered from event data with provable guarantees.

**Acknowledgement:**   Abir De is partially supported by Google India under the Google India PhD Fellowship Award, and Isabel Valera is supported by a Humboldt post-doctoral fellowship.

## Footnotes

[1] http://www.nytimes.com/2012/10/08/technology/campaigns-use-social-media-to-lure-younger-voters.html

[2] http://www.nytimes.com/2012/07/31/technology/facebook-twitter-and-foursquare-as-corporate-focus-groups.html

[3] Slant is a particular point of view from which something is seen or presented.

[4]Here, if one decides to model the message intensities with a Poisson process, $\boldsymbol{B} = 0$.

[5]For these particular networks, Poisson intensities lead to consensus while Hawkes intensities lead to polarization, however, we did find other examples in which Poisson intensities lead to polarization and Hawkes intensities lead to consensus.

[6]Here, we do not distinguish between analytical and sampling based forecasting since, in practice, they closely match each other.

[7]The failure rate is very close to zero for those datasets in which most users post messages with the same polarity.

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
