[Supplementary Material · opinion-supp.pdf]

# A Proof of Proposition 1

Given $\boldsymbol{x}^*(t) := [\boldsymbol{x}_1(t), ..., \boldsymbol{x}_{|\mathcal{V}|}(t)]^T$ and $\boldsymbol{\lambda}^*(t) := [\boldsymbol{\lambda}_1(t), ..., \boldsymbol{\lambda}_{|\mathcal{V}|}(t)]^T$, we can compactly rewrite Eqs. 3 and 4 for all users as:

$$\boldsymbol{x}^*(t) = \boldsymbol{\alpha} + \boldsymbol{A} \int_0^t g(t-\theta)\boldsymbol{m}(t) \odot d\boldsymbol{N}(t) \tag{15}$$

and

$$\boldsymbol{\lambda}^*(t) = \boldsymbol{\mu} + \boldsymbol{B} \int_0^t \kappa(t-\theta) \odot d\boldsymbol{N}(t). \tag{16}$$

Then, it easily follows that

$$d\boldsymbol{x}^*(t) = \boldsymbol{A} \int_0^t g'(t-\theta)\boldsymbol{m}(t) \odot d\boldsymbol{N}(t) + g(0)\boldsymbol{A}\boldsymbol{m}(t) \odot d\boldsymbol{N}(t), \tag{17}$$

where $g(t) = e^{-\omega t}$ and $g'(t-\theta) = -\omega g(t-\theta)$. And, we can rewrite the above expression as

$$d\boldsymbol{x}^*(t) = \omega(\boldsymbol{\alpha} - \mathbf{x}^*(t))dt + \boldsymbol{A}(\mathbf{m}(t) \odot d\boldsymbol{N}(t)). \tag{18}$$

Similarly, we can show that

$$d\boldsymbol{\lambda}^*(t) = \nu(\boldsymbol{\mu} - \boldsymbol{\lambda}^*(t))dt + \boldsymbol{B}\, d\boldsymbol{N}(t).$$

# B Auxiliary theoretical results

The proofs of Theorems 2 and 5 rely on the following auxiliary Lemma.

**Lemma 7** *The expected opinion $\mathbb{E}_{\mathcal{H}_t \backslash \mathcal{H}_{t_0}}[\boldsymbol{x}^*(t)|\mathcal{H}_{t_0}]$ in the model of opinion dynamics defined by Eqs. 4 and 3 with exponential triggering kernels with parameters $\omega$ and $\nu$ satisfies the following differential equation:*

$$\frac{d\mathbb{E}_{\mathcal{H}_t \backslash \mathcal{H}_{t_0}}[\boldsymbol{x}^*(t)|\mathcal{H}_{t_0}]}{dt} = \boldsymbol{A}\mathbb{E}_{\mathcal{H}_t \backslash \mathcal{H}_{t_0}}[\boldsymbol{x}^*(t) \odot \boldsymbol{\lambda}^*(t)|\mathcal{H}_{t_0}] - \omega\mathbb{E}_{\mathcal{H}_t \backslash \mathcal{H}_{t_0}}[\boldsymbol{x}^*(t)|\mathcal{H}_{t_0}] + \omega\boldsymbol{\alpha}, \tag{19}$$

*where $\boldsymbol{A} = (a_{vu})_{v,u \in \mathcal{G}}$ and the sign $\odot$ denotes pointwise product.*

Using that $\mathbb{E}[m_v(\theta)|x_v^*(\theta)] = x_v^*(\theta)$, we can compute the average opinion of user $u$ across all possible histories from Eq. 4 as

$$\mathbb{E}_{\mathcal{H}_t \backslash \mathcal{H}_{t_0}}[x_u^*(t)|\mathcal{H}_{t_0}] = \alpha_u + \sum_{v \in \mathcal{N}(u)} a_{uv} \int_0^t g(t-\theta)\mathbb{E}_{\mathcal{H}_t \backslash \mathcal{H}_{t_0}}[m_v(\theta)dN_v(\theta)|\mathcal{H}_{t_0}]$$

$$= \alpha_u + \sum_{v \in \mathcal{N}(u)} a_{uv} \sum_{t_i \in \mathcal{H}_v(t_0)} g(t-t_i)m_v(t_i) + \sum_{v \in \mathcal{N}(u)} a_{uv} \int_{t_0}^t g(t-\theta)\mathbb{E}_{\mathcal{H}(\theta) \backslash \mathcal{H}_{t_0}}[x_v^*(\theta)\lambda_v^*(\theta)|\mathcal{H}_{t_0}]d\theta,$$

and we can write the expectation of the opinion for all users in vectorial form as

$$\mathbb{E}_{\mathcal{H}_t \backslash \mathcal{H}_{t_0}}[\boldsymbol{x}^*(t)] = \boldsymbol{v}(t) + \boldsymbol{A} \int_0^t g(t-\theta)\mathbb{E}_{\mathcal{H}(\theta) \backslash \mathcal{H}_{t_0}}[\boldsymbol{x}^*(\theta) \odot \boldsymbol{\lambda}^*(\theta)]d\theta, \tag{20}$$

where the $\odot$ denotes pointwise product and

$$(\boldsymbol{v}(t))_u = \alpha_u + \sum_{v \in \mathcal{N}(u)} a_{uv} \sum_{t_i \in \mathcal{H}_v(t_0)} g(t-t_i)m_v(t_i).$$

Since $g(t) = e^{-\omega t}$, one may observe that $\omega\boldsymbol{v}(t) + \dot{\boldsymbol{v}}(t) = \omega\boldsymbol{\alpha}$. Then, by differentiating Eq. 20, we obtain

$$\frac{d\mathbb{E}_{\mathcal{H}_t \backslash \mathcal{H}_{t_0}}[\boldsymbol{x}^*(t)|\mathcal{H}_{t_0}]}{dt} = \boldsymbol{A}\mathbb{E}_{\mathcal{H}_t \backslash \mathcal{H}_{t_0}}[\boldsymbol{x}^*(t) \odot \boldsymbol{\lambda}^*(t)|\mathcal{H}_{t_0}] - \omega\mathbb{E}_{\mathcal{H}_t \backslash \mathcal{H}_{t_0}}[\boldsymbol{x}^*(t)|\mathcal{H}_{t_0}] + \omega\boldsymbol{\alpha}, \tag{21}$$

## C  Proof of Theorem 2

Using Lemma 7 and $\lambda_u^*(t) = \mu_u$, we obtain

$$\frac{d\mathbb{E}_{\mathcal{H}_t}[\boldsymbol{x}^*(t)]}{dt} = [-\omega I + \boldsymbol{A}\boldsymbol{\Lambda}_1]\mathbb{E}_{\mathcal{H}_t}[\boldsymbol{x}^*(t)] + \omega\boldsymbol{\alpha}, \tag{22}$$

where $\boldsymbol{\Lambda}_1 = \text{diag}[\boldsymbol{\mu}]$. Then, we apply the Laplace transform to the expression above and obtain

$$\hat{\boldsymbol{x}}(s) = [sI + \omega I - A\boldsymbol{\Lambda}_1]^{-1}\boldsymbol{x}(t_0) + \frac{\omega}{s}[sI + \omega I - A\boldsymbol{\Lambda}_1]^{-1}\boldsymbol{\alpha},$$

where we leverage the fact that, conditioning the prior history, the opinion is non-random, *i.e.*, $\mathbb{E}_{\mathcal{H}_{t_0}\backslash\mathcal{H}_{t_0^-}}[\boldsymbol{x}(t_0)|\mathcal{H}_{t_0^-}] = \boldsymbol{x}(t_0)$. Finally, applying the inverse Laplace transform, we obtain the average opinion $\mathbb{E}_{\mathcal{H}_t\backslash\mathcal{H}_{t_0}}[\boldsymbol{x}^*(t)|\mathcal{H}_{t_0}]$ in time domain as

$$\mathbb{E}_{\mathcal{H}_t\backslash\mathcal{H}_{t_0}}[\boldsymbol{x}^*(t)|\mathcal{H}_{t_0}] = e^{(\boldsymbol{A}\boldsymbol{\Lambda}_1 - \omega I)(t-t_0)}\boldsymbol{x}(t_0) + \omega(\boldsymbol{A}\boldsymbol{\Lambda}_1 - \omega I)^{-1}\left(e^{(\boldsymbol{A}\boldsymbol{\Lambda}_1 - \omega I)(t-t_0)} - \boldsymbol{I}\right)\Big]\boldsymbol{\alpha}.$$

## D  Proof of Theorem 3

Theorem 2 states that the average users' opinion $\mathbb{E}_{\mathcal{H}_t}[\boldsymbol{x}^*(t)]$ in time domain is given by

$$\mathbb{E}_{\mathcal{H}_t\backslash\mathcal{H}_{t_0}}[\boldsymbol{x}^*(t)|\mathcal{H}_{t_0}] = e^{(\boldsymbol{A}\boldsymbol{\Lambda}_1 - \omega I)(t-t_0)}\boldsymbol{x}(t_0) + \omega(\boldsymbol{A}\boldsymbol{\Lambda}_1 - \omega I)^{-1}\left(e^{(\boldsymbol{A}\boldsymbol{\Lambda}_1 - \omega I)(t-t_0)} - \boldsymbol{I}\right)\Big]\boldsymbol{\alpha}.$$

If $Re[\xi(\boldsymbol{A}\boldsymbol{\Lambda}_1)] < \omega$, where $\xi(\boldsymbol{X})$ denote the eigenvalues of matrix $\boldsymbol{X}$, it easily follows that

$$\lim_{t\to\infty}\mathbb{E}_{\mathcal{H}_t}[\boldsymbol{x}^*(t)] = \left(I - \frac{\boldsymbol{A}\boldsymbol{\Lambda}_1}{w}\right)^{-1}\boldsymbol{\alpha}. \tag{23}$$

## E  Proof of Theorem 4

Assume $b_{vu} = 0$ for all $v, u \in \mathcal{G}, v \neq u$. Then, $\lambda_v^*(t)$ only depends on user $v$'s history and, since $x_v^*(t)$ only depends on the history of the user $v$'s neighbors $\mathcal{N}(v)$, we can write

$$\mathbb{E}_{\mathcal{H}_t\backslash\mathcal{H}_{t_0}}[\boldsymbol{x}^*(t) \odot \boldsymbol{\lambda}^*(t)|\mathcal{H}_{t_0}] = \mathbb{E}_{\mathcal{H}_t\backslash\mathcal{H}_{t_0}}[\boldsymbol{x}^*(t)|\mathcal{H}_{t_0}] \odot \mathbb{E}_{\mathcal{H}_t\backslash\mathcal{H}_{t_0}}[\boldsymbol{\lambda}^*(t)],$$

and rewrite Eq. 21 as

$$\frac{d\mathbb{E}_{\mathcal{H}_t\backslash\mathcal{H}_{t_0}}[\boldsymbol{x}^*(t)|\mathcal{H}_{t_0}]}{dt} = \tag{24}$$
$$\boldsymbol{A}(\mathbb{E}_{\mathcal{H}_t\backslash\mathcal{H}_{t_0}}[\boldsymbol{x}^*(t)|\mathcal{H}_{t_0}] \odot \mathbb{E}_{\mathcal{H}_t\backslash\mathcal{H}_{t_0}}[\boldsymbol{\lambda}^*(t)|\mathcal{H}_{t_0}]) - \omega\mathbb{E}_{\mathcal{H}_t\backslash\mathcal{H}_{t_0}}[\boldsymbol{x}^*(t)|\mathcal{H}_{t_0}] + \omega\boldsymbol{\alpha}.$$

We can now compute $\mathbb{E}_{\mathcal{H}(\theta)\backslash\mathcal{H}_{t_0}}[\boldsymbol{\lambda}^*(\theta)|\mathcal{H}_{t_0}]$ analytically as follows. From Eq. 3, we obtain

$$\mathbb{E}_{\mathcal{H}_t\backslash\mathcal{H}_{t_0}}[\boldsymbol{\lambda}^*(t)|\mathcal{H}_{t_0}] = \boldsymbol{\eta}(t) + \int_{t_0}^t \boldsymbol{B}\kappa(t-\theta)\mathbb{E}_{\mathcal{H}_\theta\backslash\mathcal{H}_{t_0}}[\boldsymbol{\lambda}^*(\theta)]d\theta, \tag{25}$$

where $\kappa(t) = e^{-\nu t}$, $[\boldsymbol{\eta}(t)]_{u\in\mathcal{V}} = \mu_u + \sum_{v\in\mathcal{N}(u)} b_{uv} \sum_{t_i\in\mathcal{H}_v(t_0)} \kappa(t - t_i)$ and $\boldsymbol{B} = (b_{vu})_{v,u\in\mathcal{V}}$, where $b_{vu} = 0$ for all $v \neq u$, by assumption. Differentiating with respect to $t$, we get,

$$\frac{d}{dt}\mathbb{E}_{\mathcal{H}_t\backslash\mathcal{H}_{t_0}}[\boldsymbol{\lambda}^*(t)|\mathcal{H}_{t_0}] = -\nu\mathbb{E}_{\mathcal{H}_t\backslash\mathcal{H}_{t_0}}[\boldsymbol{\lambda}^*(t)|\mathcal{H}_{t_0}] + \boldsymbol{B}\mathbb{E}_{\mathcal{H}_t\backslash\mathcal{H}_{t_0}}[\boldsymbol{\lambda}^*(t)|\mathcal{H}_{t_0}] + \nu\boldsymbol{\mu},$$

with initial condition $\mathbb{E}_{\mathcal{H}(t_0^+)\backslash\mathcal{H}(t_0)}\boldsymbol{\lambda}^*(t_0) = \boldsymbol{\eta}(t_0)$. By taking the Laplace transform and then applying inverse Laplace transform,

$$\mathbb{E}_{\mathcal{H}_t\backslash\mathcal{H}_{t_0}}[\boldsymbol{\lambda}^*(t)|\mathcal{H}_{t_0}] = e^{(\boldsymbol{B}-\nu I)(t-t_0)}\boldsymbol{\eta}(t_0) + \nu(\boldsymbol{B}-\nu I)^{-1}\left(e^{(\boldsymbol{B}-\nu I)(t-t_0)} - \boldsymbol{I}\right)\boldsymbol{\mu} \; \forall t \geq t_0, \tag{26}$$

where $\boldsymbol{\eta}(t_0) = \mathbb{E}_{\mathcal{H}(t_0^+)\backslash\mathcal{H}(t_0)}[\boldsymbol{\lambda}^*(t_0)]$. Using Eqs. 24 and 26, as well as $\mathbb{E}_{\mathcal{H}_t\backslash\mathcal{H}_{t_0}}[\boldsymbol{x}^*(t)] \odot \mathbb{E}_{\mathcal{H}_t\backslash\mathcal{H}_{t_0}}[\boldsymbol{\lambda}^*(t)] = \boldsymbol{\Lambda}(t)\mathbb{E}_{\mathcal{H}_t\backslash\mathcal{H}_{t_0}}[\boldsymbol{x}^*(t)]$, where $\boldsymbol{\Lambda}(t) := \text{diag}[\mathbb{E}_{\mathcal{H}_t\backslash\mathcal{H}_{t_0}}[\boldsymbol{\lambda}^*(t)]]$, we obtain

$$\frac{d\mathbb{E}_{\mathcal{H}_t\backslash\mathcal{H}_{t_0}}[\boldsymbol{x}^*(t)]}{dt} = [-\omega I + \boldsymbol{A}\boldsymbol{\Lambda}(t)]\mathbb{E}_{\mathcal{H}_t\backslash\mathcal{H}_{t_0}}[\boldsymbol{x}^*(t)] + \omega\boldsymbol{\alpha},$$

with initial conditions $(\mathbb{E}_{\mathcal{H}(t_0^+)\backslash\mathcal{H}(t_0)}[\boldsymbol{x}^*(t_0)])_{u\in\mathcal{V}} = \alpha_u + \sum_{v\in\mathcal{N}(u)} a_{uv} \sum_{t_i\in\mathcal{H}_v(t_0)} g(t_0 - t_i)m_v(t_i)$.

# F    Proof of Theorem 5

Theorem 4 states that the average users' opinion $\mathbb{E}_{\mathcal{H}_t}[\boldsymbol{x}^*(t)]$ in time domain is given by

$$\frac{d\mathbb{E}_{\mathcal{H}_t \setminus \mathcal{H}_{t_0}}[\boldsymbol{x}^*(t)|\mathcal{H}_{t_0}]}{dt} = [-\omega I + \boldsymbol{A}\boldsymbol{\Lambda}(t)]\mathbb{E}_{\mathcal{H}_t \setminus \mathcal{H}_{t_0}}[\boldsymbol{x}^*(t)|\mathcal{H}_{t_0}] + \omega\boldsymbol{\alpha}. \tag{27}$$

In such systems, solutions can be written as [16]

$$\mathbb{E}_{\mathcal{H}_t \setminus \mathcal{H}_{t_0}}[\boldsymbol{x}^*(t)|\mathcal{H}_{t_0}] = \Phi(t)\alpha + \omega \int_0^t \Phi(s)\boldsymbol{\alpha}ds, \tag{28}$$

where the transition matrix $\Phi(t)$ defines as a solution of the matrix differential equation

$$\dot{\Phi}(t) = [-\omega I + \boldsymbol{A}\boldsymbol{\Lambda}(t)]\Phi(t) \text{ with } \Phi(0) = I.$$

If $\Phi(t)$ satisfies $||\Phi(t)|| \leq \gamma e^{-ct} \, \forall t > 0$ for $\gamma, c > 0$ then the steady state solution to Eq. 28 is given by [16]

$$\lim_{t \to \infty} \mathbb{E}_{\mathcal{H}_t \setminus \mathcal{H}_{t_0}}[\boldsymbol{x}^*(t)|\mathcal{H}_{t_0}] = \left(I - \frac{\boldsymbol{A}\boldsymbol{\Lambda}_2}{\omega}\right)^{-1}\boldsymbol{\alpha}.$$

where $\boldsymbol{\Lambda}_2 = \lim_{t \to \infty} \boldsymbol{\Lambda}(t) = \text{diag}\left[I - \frac{B}{\nu}\right]^{-1}\boldsymbol{\mu}$.

# G    Proof of Theorem 6

Let $\{\boldsymbol{x}_l^*(t)\}_{l=1}^n$ be the simulated opinions for all users and define $\boldsymbol{s}(t) = \frac{1}{n}\sum_{l=1}^n \boldsymbol{s}_l(t)$, where $\boldsymbol{s}_l(t) = (\hat{\boldsymbol{x}}_l^*(t) - \mathbb{E}_{\mathcal{H}_t \setminus \mathcal{H}_{t_0}}[\boldsymbol{x}^*(t)|\mathcal{H}_{t_0}])$. Clearly, for a given $t$, all elements in $\boldsymbol{s}_l(t)$ are i.i.d. random variables with zero mean and variance, and we can bound $|s_u(t)| < 2x_{\max}$. Then, by Bernstein's inequality, the following holds true,

$$\mathbb{P}(|s_u(t)| > \epsilon) = \mathbb{P}(s_u(t) > \epsilon) + \mathbb{P}(s_u(t) < -\epsilon) > 2.\exp\left(-\frac{3n\epsilon^2}{6\sigma_{\mathcal{H}_t \setminus \mathcal{H}_{t_0}}^2(x_u^*(t)|\mathcal{H}_{t_0}) + 4x_{\max}\epsilon}\right)$$

Let $\sigma_{\max}^2(t) = \max_{u \in \mathcal{G}} \sigma_{\mathcal{H}_t \setminus \mathcal{H}_{t_0}}^2(x_u^*(t)|\mathcal{H}_{t_0})$. If we choose,

$$\delta < 2.\exp\left(-\frac{3n\epsilon^2}{6\sigma_{\max}^2(t) + 4x_{\max}\epsilon}\right) \tag{29}$$

we obtain the required bound for $n$. Moreover, given this choice of $\delta$, we have $\mathbb{P}(|s_u(t)| < \epsilon) > 1 - \delta$ immediately.

However, a finite bound on $n$ requires the variance $\sigma_{\max}^2(t)$ to be bounded for all $t$. Hence, we analyze the variance and its stability below.

## G.1    Dynamics of variance

In this section we compute the time-domain evolution of the variance and characterize its stability for Poisson driven opinion dynamics. A general analysis of the variance for multidimensional Hawkes is left for future work.

**Lemma 8** *Given a collection of messages $\mathcal{H}_{t_0}$ recorded during a time period $[0, t_0)$ and $\lambda_u^*(t) = \mu_u$ for all $u \in \mathcal{G}$, the covariance matrix $\boldsymbol{\Gamma}(t_0, t)$ at any time $t$ conditioned on the history $\mathcal{H}_{t_0}$ can be described as,*

$$vec(\boldsymbol{\Gamma}(t_0, t)) = \int_0^t \boldsymbol{\Phi}(t - \theta)vec[\sigma^2 \boldsymbol{A}\boldsymbol{\Lambda}\boldsymbol{A}^T + \boldsymbol{A}\,\text{diag}(\mathbb{E}_{\mathcal{H}_{t^-}}[\boldsymbol{x}^*(\theta)])^2\boldsymbol{\Lambda}\boldsymbol{A}^T]d\theta.$$

*where*

$$\boldsymbol{\Phi}(t) = e^{\left((-\omega I + \boldsymbol{A}\boldsymbol{\Lambda})\otimes I + I \otimes(-\omega I + \boldsymbol{A}\boldsymbol{\Lambda}) + (\boldsymbol{A}\otimes\boldsymbol{A})\widehat{\boldsymbol{\Lambda}}\right)t},$$

$\hat{\boldsymbol{\Lambda}}_{i^2, i^2} = \boldsymbol{\lambda}^*(i)$, *and* $\boldsymbol{\Lambda} := \text{diag}[\boldsymbol{\lambda}]$. *Moreover, the stability of the system is characterized by*

$$\xi\left[(-\omega I + \boldsymbol{A}\boldsymbol{\Lambda}) \otimes I + I \otimes (-\omega I + \boldsymbol{A}\boldsymbol{\Lambda}) + (\boldsymbol{A} \otimes \boldsymbol{A})\widehat{\boldsymbol{\Lambda}}\right] < 0.$$

**Proof.** By definition, the covariance matrix is given by

$$\mathbf{\Gamma}(t_0, t) := \mathbb{E}_{\mathcal{H}_t \setminus \mathcal{H}_{t_0}}[(\boldsymbol{x}^*(t) - \mathbb{E}_{\mathcal{H}_t \setminus \mathcal{H}_{t_0}}(\boldsymbol{x}^*(t)))(\boldsymbol{x}^*(t) - \mathbb{E}_{\mathcal{H}_t \setminus \mathcal{H}_{t_0}}(\boldsymbol{x}^*(t)))^T | \mathcal{H}_{t_0}]. \tag{30}$$

Hence, if we define $\Delta \boldsymbol{x} = (\boldsymbol{x}^*(t) - \mathbb{E}_{\mathcal{H}_t \setminus \mathcal{H}_{t_0}}(\boldsymbol{x}^*(t)))$, we can compute the differential of the covariance matrix as

$$d\mathbf{\Gamma}(t_0, t) = \mathbb{E}_{\mathcal{H}_t \setminus \mathcal{H}_{t_0}}[d(\Delta \boldsymbol{x} \Delta \boldsymbol{x}^T) | \mathcal{H}_{t_0}] = \mathbb{E}_{\mathcal{H}_t \setminus \mathcal{H}_{t_0}}[\Delta \boldsymbol{x} d(\Delta \boldsymbol{x}^T) + d(\Delta \boldsymbol{x}) \Delta \boldsymbol{x}^T + d(\Delta \boldsymbol{x}) d(\Delta \boldsymbol{x}^T) | \mathcal{H}_{t_0}], \tag{31}$$

where

$$d(\Delta \boldsymbol{x}) = d(\boldsymbol{x}^*(t) - \mathbb{E}_{\mathcal{H}_t \setminus \mathcal{H}_{t_0}}(\boldsymbol{x}^*(t))) = d(\boldsymbol{x}^*(t)) - d(\mathbb{E}_{\mathcal{H}_t \setminus \mathcal{H}_{t_0}}(\boldsymbol{x}^*(t))). \tag{32}$$

Next, note that

$$\mathbb{E}(d\boldsymbol{N}(t)d\boldsymbol{N}^T(t)) = \mathbb{E}[dN_i(t)dN_j(t)]_{i,j \in \mathcal{V}} = \mathbb{E}[\text{diag}(d\boldsymbol{N}(t))] = \mathbf{\Lambda}, \tag{33}$$

where the off-diagonal entries vanish, since two jumps cannot happen at the same time point [14]. Now, recall the Markov representation of our model, *i.e.*,

$$d\boldsymbol{x}^*(t) = -\omega \boldsymbol{x}^*(t)dt + \boldsymbol{A}\boldsymbol{M}^*(t)d\boldsymbol{N}(t) + \boldsymbol{\alpha}dt, \tag{34}$$

where $\boldsymbol{M}^*(t) := \text{diag}[\boldsymbol{m}(t)]$ is the diagonal formed by the sentiment vector and note that,

$$\boldsymbol{m}(t) \odot d\boldsymbol{N}(t) = \boldsymbol{M}^*(t)d\boldsymbol{N}(t) = \text{diag}[d\boldsymbol{N}(t)]\boldsymbol{m}(t), \tag{35}$$

and, using Eq. 21,

$$d\mathbb{E}_{\mathcal{H}_t \setminus \mathcal{H}_{t_0}}[\boldsymbol{x}^*(t) | \mathcal{H}_{t_0}] = -\omega \mathbb{E}_{\mathcal{H}_t \setminus \mathcal{H}_{t_0}}[\boldsymbol{x}^*(t) | \mathcal{H}_{t_0}]dt + \boldsymbol{A}\mathbf{\Lambda}\mathbb{E}_{\mathcal{H}_t \setminus \mathcal{H}_{t_0}}[\boldsymbol{x}^*(t) | \mathcal{H}_{t_0}]dt + \omega \boldsymbol{\alpha}dt. \tag{36}$$

Then, if we substitute Eqs. 34 and 36 in Eq. 32, we obtain

$$d(\Delta \boldsymbol{x}) = -\omega [\boldsymbol{x}^*(t) - \mathbb{E}_{\mathcal{H}_t \setminus \mathcal{H}_{t_0}}(\boldsymbol{x}^*(t) | \mathcal{H}_{t_0})]dt + \boldsymbol{A}[\boldsymbol{M}^*(t)d\boldsymbol{N}(t) - \mathbf{\Lambda}\mathbb{E}_{\mathcal{H}_t \setminus \mathcal{H}_{t_0}}(\boldsymbol{x}^*(t) | \mathcal{H}_{t_0})dt]. \tag{37}$$

As a result, we can write

$$d\mathbf{\Gamma}(t_0, t) = \mathbb{E}_{\mathcal{H}_t \setminus \mathcal{H}_{t_0}}\Big[ -2\omega \Delta \boldsymbol{x} \Delta \boldsymbol{x}^T dt + \underbrace{\Delta x((\boldsymbol{M}^*(t)d\boldsymbol{N}(t) - \mathbf{\Lambda}\mathbb{E}_{\mathcal{H}_t \setminus \mathcal{H}_{t_0}}(\boldsymbol{x}^*(t))dt)^T \boldsymbol{A}^T}_{\text{Term 1}}$$

$$+ \boldsymbol{A}(\boldsymbol{M}^*(t)d\boldsymbol{N}(t) - \mathbf{\Lambda}\mathbb{E}_{\mathcal{H}_t \setminus \mathcal{H}_{t_0}}(\boldsymbol{x}^*(t))dt)\Delta x^T$$

$$+ \underbrace{\boldsymbol{A}(\boldsymbol{M}^*(t)d\boldsymbol{N}(t) - \mathbf{\Lambda}\mathbb{E}_{\mathcal{H}_t \setminus \mathcal{H}_{t_0}}(\boldsymbol{x}^*(t))dt)(\boldsymbol{M}^*(t)d\boldsymbol{N}(t) - \mathbf{\Lambda}\mathbb{E}_{\mathcal{H}_t \setminus \mathcal{H}_{t_0}}(\boldsymbol{x}^*(t))dt)^T \boldsymbol{A}^T}_{\text{Term 2}} | \mathcal{H}_{t_0}\Big], \tag{38}$$

where Term 1 gives

$$\mathbb{E}_{\mathcal{H}_t \setminus \mathcal{H}_{t_0}}\big[\Delta x(\boldsymbol{M}^*(t)d\boldsymbol{N}(t) - \mathbf{\Lambda}\mathbb{E}_{\mathcal{H}_t \setminus \mathcal{H}_{t_0}}(\boldsymbol{x}^*(t))dt)^T \boldsymbol{A}^T | \mathcal{H}_{t_0}\big]$$

$$= \mathbb{E}_{\mathcal{H}_t \setminus \mathcal{H}_{t_0}}\big[\Delta x(m^*(t) - \mathbb{E}_{\mathcal{H}_t \setminus \mathcal{H}_{t_0}}(\boldsymbol{x}^*(t))dt)^T \boldsymbol{A}^T | \mathcal{H}_{t_0}\big]\mathbf{\Lambda}dt \quad \text{(Using Eq. 35 and the fact that } \mathbb{E}[\text{diag}(d\boldsymbol{N}(t))] = \mathbf{\Lambda})$$

$$= \mathbb{E}_{\mathcal{H}_t \setminus \mathcal{H}_{t_0}}\Big[\Delta x.\mathbb{E}\big[(m^*(t) - \mathbb{E}_{\mathcal{H}_t \setminus \mathcal{H}_{t_0}}(\boldsymbol{x}^*(t)))^T \boldsymbol{A}^T | x(t), \mathcal{H}_{t_0}\big]\Big]\mathbf{\Lambda}dt$$

$$= \mathbf{\Gamma}(t_0, t)\mathbf{\Lambda}dt,$$

and Term 2 gives

$$\mathbb{E}_{\mathcal{H}_t \setminus \mathcal{H}_{t_0}}\boldsymbol{A}[(\boldsymbol{M}^*(t)d\boldsymbol{N}(t) - \mathbf{\Lambda}\mathbb{E}_{\mathcal{H}_t \setminus \mathcal{H}_{t_0}}(\boldsymbol{x}^*(t))dt)(\boldsymbol{M}^*(t)d\boldsymbol{N}(t) - \mathbf{\Lambda}\mathbb{E}_{\mathcal{H}_t \setminus \mathcal{H}_{t_0}}(\boldsymbol{x}^*(t))dt)^T | \mathcal{H}_{t_0}]\boldsymbol{A}^T$$

$$= \boldsymbol{A}\mathbb{E}_{\mathcal{H}_t \setminus \mathcal{H}_{t_0}}[\boldsymbol{M}^*(t)\mathbb{E}(d\boldsymbol{N}(t)d\boldsymbol{N}^T(t))\boldsymbol{M}^*(t) | \mathcal{H}_{t_0}]\boldsymbol{A}^T + O(dt^2)$$

$$= \boldsymbol{A}\mathbb{E}_{\mathcal{H}_t \setminus \mathcal{H}_{t_0}}[\boldsymbol{M}^2(t) | \mathcal{H}_{t_0}]\mathbf{\Lambda}dt\boldsymbol{A}^T \quad \text{(From Eq. 33)}$$

$$= \boldsymbol{A}(\sigma^2 I + \text{diag}(\mathbb{E}_{\mathcal{H}_t \setminus \mathcal{H}_{t_0}}(x(t)^2) | \mathcal{H}_{t_0}))\mathbf{\Lambda}\boldsymbol{A}^T dt$$

$$= \boldsymbol{A}\big[\sigma^2 I + \mathbf{\Gamma}_{ii}(t_0, t) + \text{diag}[\mathbb{E}_{\mathcal{H}_t \setminus \mathcal{H}_{t_0}}(\boldsymbol{x}^*(t) | \mathcal{H}_{t_0})]^2)\big]\mathbf{\Lambda}\boldsymbol{A}^T dt.$$

Hence, substituting the expectations above into Eq. 38, we obtain

$$\frac{d\mathbf{\Gamma}(t_0, t)}{dt} = -2\omega \mathbf{\Gamma}(t_0, t) + \mathbf{\Gamma}(t_0, t)\mathbf{\Lambda}\boldsymbol{A}^T + \boldsymbol{A}\mathbf{\Lambda}\mathbf{\Gamma}(t_0, t) \tag{39}$$

$$+ \boldsymbol{A}\mathbf{\Gamma}_{ii}(t)\mathbf{\Lambda}\boldsymbol{A}^T + \sigma^2 \boldsymbol{A}\mathbf{\Lambda}\boldsymbol{A}^T + \boldsymbol{A}\text{diag}(\mathbb{E}_{\mathcal{H}_t \setminus \mathcal{H}_{t_0}}[\boldsymbol{x}^*(t) | \mathcal{H}_{t_0}])^2 \mathbf{\Lambda}\boldsymbol{A}^T,$$

Eq. 39 can be readily written in vectorial form by exploiting the properties of the Kronecker product as

$$\frac{d[\text{vec}(\boldsymbol{\Gamma}(t_0, t)])}{dt} = \boldsymbol{V}(t)\text{vec}(\boldsymbol{\Gamma}(t_0, t)) + \text{vec}\big(\sigma^2 \boldsymbol{A\Lambda A}^T + \boldsymbol{A}\,\text{diag}(\mathbb{E}_{\mathcal{H}_t \backslash \mathcal{H}_{t_0}}[\boldsymbol{x}^*(t)|\mathcal{H}_{t_0}])^2 \boldsymbol{\Lambda A}^T\big)$$

where

$$\boldsymbol{V}(t) = (-\omega I + \boldsymbol{A\Lambda}) \otimes I + I \otimes (-\omega I + \boldsymbol{A\Lambda}) + \sum_{i=1}^{n} (\boldsymbol{A\Lambda} P_i \otimes A P_i),$$

$P_i = [\delta_{ii}]$ and $\otimes$ stands for the Kronecker product. Hence, the closed form solution of the above equation can be written as,

$$\text{vec}(\boldsymbol{\Gamma}(t_0, t)) = \int_0^t e^{\boldsymbol{V}(t-\theta)} \text{vec}[\sigma^2 \boldsymbol{A\Lambda A}^T + \boldsymbol{A}\,\text{diag}(\mathbb{E}_{\mathcal{H}(\theta)\backslash \mathcal{H}_{t_0}}[\boldsymbol{x}^*(\theta)|\mathcal{H}_{t_0}])^2 \boldsymbol{\Lambda A}^T]d\theta. \quad (40)$$

Moreover, the covariance matrix $\boldsymbol{\Gamma}(t_0, t)$ is bounded iff

$$\text{Re}\big[\lambda[(-\omega I + \boldsymbol{A\Lambda}) \otimes I + I \otimes (-\omega I + \boldsymbol{A\Lambda}) + (\boldsymbol{A} \otimes \boldsymbol{A})\widehat{\boldsymbol{\Lambda}}]\big] < 0,$$

where $\widehat{\boldsymbol{\Lambda}} := \sum_{i=1}^{|\mathcal{V}|} \boldsymbol{\Lambda} P_i \otimes P_i$.

In this case, the steady state solution $\boldsymbol{\Gamma} = \lim_{t \to \infty} \boldsymbol{\Gamma}(t_0, t)$ is given by

$$-2\omega \boldsymbol{\Gamma} + \boldsymbol{\Gamma \Lambda A}^T + \boldsymbol{A\Lambda \Gamma} + \boldsymbol{A}\,\text{diag}(\boldsymbol{\Gamma})\boldsymbol{\Lambda A}^T + \sigma^2 \boldsymbol{A\Lambda A} + \boldsymbol{A}\,\text{diag}(\mathbb{E}_{\mathcal{H}_{t_-}}[\boldsymbol{x}^*])^2 \boldsymbol{\Lambda A}^T = 0. \quad (41)$$

that means $\lim_{t \to \infty} \text{vec}[\boldsymbol{\Gamma}(t_0, t)]$ is same as,

$$\text{vec}(\boldsymbol{\Gamma}) = [(-\omega I + \boldsymbol{A\Lambda}) \otimes I + I \otimes (-\omega I + \boldsymbol{A\Lambda})^T + (\boldsymbol{A} \otimes \boldsymbol{A})\widehat{\boldsymbol{\Lambda}}]^{-1}$$
$$\times \text{vec}[\sigma^2 \boldsymbol{A\Lambda A}^T + \boldsymbol{A}\,\text{diag}(\boldsymbol{x}_\infty^*)^2 \boldsymbol{\Lambda A}^T] \quad (42)$$

where $\boldsymbol{x}^* = \lim_{t \to \infty} \mathbb{E}_{\mathcal{H}_t \backslash \mathcal{H}_{t_0}}(\boldsymbol{x}^*(t)|\mathcal{H}_{t_0})$. Finally, the variance of node $u$'s opinion at any time $t$ is given by

$$\sigma^2_{\mathcal{H}_t \backslash \mathcal{H}_{t_0}}(x_u^*(t)|\mathcal{H}_{t_0}) = [\boldsymbol{\Gamma}(t_0, t)]_{u,u} = \text{vec}(P_u)^T \text{vec}(\boldsymbol{\Gamma}(t_0, t)),$$

where $P_u$ is the sparse matrix with its only $(u, u)$ entry to be equal to unity.

# H  Parameter estimation algorithm

## H.1  Estimation of $\alpha$ and $A$

Algorithm H.1 summarizes the estimation of the opinion dynamics, *i.e.*, $\alpha$ and $A$, which reduces to a least-square problem.

---

**Algorithm 1:** Estimation of $\alpha$ and $A$

---

1: **Input:** $\mathcal{H}_T, G(V,E)$ , regularizer $\lambda$, error bound $\epsilon$
2: **Output:** $\hat{\alpha}$, $\hat{A}$
3: **Initialize:**
4: IndexForV$[1:|V|] = \vec{0}$
5: **for** $i=1$ to $|\mathcal{H}(T)|$ **do**
6:     $\mathcal{T}[u_i](\text{IndexForV}[u_i]) = (t_i, u_i, m_i)$
7:     IndexForV$[u_i]$++
8: **end for**
9: **for** $u \in \mathcal{V}$ **do**
10:     $i = 0$
11:     $S = \mathcal{T}[u]$
12:     **for** $v \in \mathcal{N}(u)$ **do**
13:         $S =$MergeTimes$(S, \mathcal{T}[v])$
14:         `// Merges two sets w.r.t` $t_i$`, takes` $O(|\mathcal{H}_u(T)| + |\cup_{v\in\mathcal{N}(u)} \mathcal{H}_v(T)|)$ `steps`
15:         $x_{\text{last}} = 0$
16:         $t_{\text{last}} = 0$
17:         $j = 0$
18:         **for** $i = 1$ to $|S|$ **do**
19:             `// This loop takes` $O(|S|) = O(|\mathcal{H}_u(T)| + |\cup_{v\in\mathcal{N}(u)} \mathcal{H}_v(T)|)$ `steps`
20:             $(t_v, v, m_v) = S[i]$
21:             $t_{\text{now}} = t_v$
22:             **if** $u = v$ **then**
23:                 $x_{\text{now}} = x_{\text{last}}e^{-\omega(t_{\text{now}} - t_{\text{last}})}$
24:                 $j$++
25:                 $g[u](j, v) = x_{\text{now}}$
26:                 $Y[u](j) = m_u$
27:             **else**
28:                 $x_{\text{now}} = x_{\text{last}}e^{-\omega(t_{\text{now}} - t_{\text{last}})} + m_v$
29:             **end if**
30:             $t_{\text{last}} = t_{\text{now}}$
31:             $x_{\text{last}} = x_{\text{now}}$
32:         **end for**
33:     **end for**
34: **end for**
35: `// Estimation of` $(\alpha, A)$
36: **for** $u \in \mathcal{V}$ **do**
37:     $a =$InferOpinionParams$(Y[u], \lambda, g[u])$
38:     $\hat{\alpha}_u = a[1]$
39:     $\hat{A}[*][u] = a[1 : \text{end}]$
40: **end for**

---

**Algorithm 2:** InferOpinionParams($Y_u, \lambda, g_u$)

---

1: $s = \text{numRows}(g_u)$
2: $X = [\vec{\mathbf{1}}_s \; g_u]$
3: $Y = Y_u$
4: $L = X'X$
5: $x = (\lambda I + L)^{-1} X'Y$
6: return $x$

---

## H.2 Estimation of $(\boldsymbol{\mu}, \boldsymbol{B})$

The first step of the parameter estimation procedure for temporal dynamics also involves the computation of the triggering kernels, which we do in same way as for the opinion dynamics in Algorithm H.1. In order to estimate the parameters, we adopted the spectral projected gradient descent (SPG) method proposed in [4].

---

**Algorithm 3:** Spectral projected gradient descent for $\mu_u$ and $\mathbf{B}_{u*}$

---

1: **Input:** $\mathcal{H}_T$, $G(V,E)$ and $\mu_u^0, \mathbf{B}_{u*}^0$, step-length bound $\alpha_{max} > 0$ and initial-step length $\alpha_{bb} \in (0, \alpha_{max}]$, error bound $\epsilon$ and size of memory $h$
2: **Output:** $\hat{\mu}_u, \hat{\mathbf{B}}_{u*}$
3: **Notation:**
  $x = [\mu_u, \mathbf{B}_{u*}]$
  $f(x) = \sum_{e_i \in \mathcal{H}(T)} \log \lambda_{u_i}^*(t_i) - \sum_{u \in \mathcal{V}} \int_0^T \lambda_u^*(\tau)\, d\tau$
4: **Initialize:**
  $k = 0$
  $x_0 = [\mu_u^0, \mathbf{B}_{u*}^0]$
5: **while** $||d_k|| < \epsilon$ **do**
6:     $\bar{\alpha}_k \leftarrow \min\{\alpha_{\max}, \alpha_{bb}\}$
7:     $d_k \leftarrow P_c[x_k - \bar{\alpha}_k \nabla f^u(x_k)] - x_k$
8:     $f_b^u \leftarrow \max\{f^u(x_k), f^u(x_{k-1}), ..., f^u(x_{k-h})\}$
9:     $\alpha \leftarrow 1$
10:     **while** $q_k(x_k + \alpha d_k) > f_b^u + \nu \alpha \nabla f^u(x_k)^T d_k$ **do**
11:        Select $\alpha$ by backtracking line-search;
12:     **end while**
13:     $x_{k+1} \leftarrow x_k + \alpha d_k$
14:     $s_k \leftarrow \alpha d_k$
15:     $y_k \leftarrow \alpha B_k d_k$
16:     $\alpha_{bb} \leftarrow y_k^T y_k / s_k^T y_k$
17:     $k = k + 1$
18: **end while**

---

# I Model simulation algorithm

We leverage the simulation algorithm in [11] to design an efficient algorithm for simulating opinion-dynamics. The two basic premises of the simulation algorithm are the sparsity of the network and the Markovian property of the model. Due to sparsity, any sampled event would effect only a few number of intensity functions, only those of the local neighbors of the node. Therefore, to generate the new sample and to identify the intensity functions that require changes, we need $O(\log |\mathcal{V}|)$ operations to maintain the heap for the priority queue. On the other hand, the Markovian property allows us to update the rate and opinion in $O(1)$ operations. The worst-case time-complexity of this algorithm can be found to be $O(d_{\max} |\mathcal{H}(T)||\mathcal{V}|)$, where $d_{\max}$ is the maximum degree.

---
**Algorithm 4:** OpinionModelSimulation($T, \boldsymbol{\mu}, \boldsymbol{B}, \boldsymbol{\alpha}, \boldsymbol{A}$)
---
1: Initialize the priority queue $Q$
2: LastOpinionUpdateTime$[1:|V|]$ = LastIntensityUpdateTime$[1:|V|]$ = $\vec{0}$
3: LastOpinionUpdateValue$[1:|V|]$ = $\boldsymbol{\alpha}$
4: LastIntensityUpdateValue$[1:|V|]$ = $\boldsymbol{\mu}$
5: $\mathcal{H}(0) \leftarrow \emptyset$
6: **for** $u \in \mathcal{V}$ **do**
7:     $t =$SampleEvent$(\boldsymbol{\mu}[u], 0, u)$
8:     $Q$.insert$([t, u])$
9: **end for**
10: **while** $t < T$ **do**
11:     `%%Approve the minimum time of all posts`
12:     $[t', u] = Q$.ExtractMin()
13:     $t_u =$ LastOpinionUpdateTime$[u]$
14:     $x_{t_u} =$ LastOpinionUpdateValue$[u]$
15:     $x_{t'_u} \leftarrow \boldsymbol{\alpha}[u] + (x_{t_u} - \boldsymbol{\alpha}[u])e^{-\omega(t - t_u)}$
16:     LastOpinionUpdateTime$[u] = t'$
17:     LastOpinionUpdateValue$[u] = x_{t'_u}$
18:     $m_u \sim p(\boldsymbol{m}|x_u(t))$
19:     $\mathcal{H}(t') \leftarrow \mathcal{H}(t) \cup (t, m_u, u)$
20:     `%% Update neighbors'`
21:     **for** $\forall v$ such that $u \rightsquigarrow v$ **do**
22:         $t_v =$ LastIntensityUpdateTime$[v]$
23:         $\lambda_{t_v} =$ LastIntensityUpdateValue$[v]$
24:         $\lambda_{t'_v} \leftarrow \boldsymbol{\mu}[v] + (\lambda_{t_v} - \boldsymbol{\mu}[v])e^{-\omega(t - t_v)} + \boldsymbol{B}_{uv}$
25:         LastIntensityUpdateTime$[v] = t'$
26:         LastIntensityUpdateValue$[v] = \lambda_{t'_v}$
27:         $t_v =$ LastOpinionUpdateTime$[v]$
28:         $x_{t_v} =$ LastOpinionUpdateValue$[v]$
29:         $x_{t'_v} \leftarrow \boldsymbol{\alpha}[v] + (x_{t_v} - \boldsymbol{\alpha}[v])e^{-\omega(t - t_v)} + \boldsymbol{A}_{uv}m_u$
30:         LastOpinionUpdateTime$[v] = t'$
31:         LastOpinionUpdateValue$[v] = x_{t'_v}$
32:         `%%Sample by only effected nodes`
33:         $t_+ =$SampleEvent$(\lambda_v(t'), t', v)$
34:         $Q$.UpdateKey$(v, t_+)$
35:     **end for**
36:     $t \leftarrow t'$
37: **end while**
38: return $\mathcal{H}(T)$
---

---
**Algorithm 5:** SampleEvent($\lambda, t, v$)
---
1: $\bar{\lambda} = \lambda$, $s \leftarrow t$
2: **while** $s < T$ **do**
3:     $\mathcal{U} \sim$ Uniform$[0, 1]$
4:     $s = s - \frac{\ln \mathcal{U}}{\bar{\lambda}}$
5:     $\lambda(s) \leftarrow \boldsymbol{\mu}[v] + (\lambda_v(t) - \boldsymbol{\mu}[v])e^{-\omega(s - t)}$
6:     $d \sim$ Uniform$[0, 1]$
7:     **if** $d\bar{\lambda} < \lambda$ **then**
8:         break
9:     **else**
10:         $\bar{\lambda} = \lambda(s)$
11:     **end if**
12: **end while**
13: return $s$
---

## J    Experiments on Synthetic Data

**Parameter estimation accuracy.** We evaluate the accuracy of our model estimation procedure on three types of Kronecker networks [20]: i) Assortative networks (parameter matrix $[0.96, 0.3; 0.3, 0.96]$), in which nodes tend to link to nodes with similar degree; ii) dissortative networks ($[0.3, 0.96; 0.96, 0.3]$), in which nodes tend to link to nodes with different degree; and iii) core-periphery networks ($[0.9, 0.5; 0.5, 0.3]$). For each network, the message intensities are multivariate Hawkes, $\boldsymbol{\mu}$ and $\boldsymbol{B}$ are drawn from a uniform distribution $U(0,1)$, and $\boldsymbol{\alpha}$ and $\boldsymbol{A}$ are drawn from a Gaussian distribution $\mathcal{N}(\mu = 0, \sigma = 1)$. We use exponential kernels with parameters $\omega = 100$ and $\nu = 1$, respectively, for opinions $\mathbf{x}_u^*(t)$ and intensities $\boldsymbol{\lambda}^*(t)$. We evaluate the accuracy of our estimation procedure in terms of mean squared error (MSE), between the estimated and true parameters, *i.e.*, $\mathbb{E}[(x - \hat{x})^2]$. Figure 5 shows the MSE of the parameters $(\boldsymbol{\alpha}, \boldsymbol{A})$, which control the Hawkes intensities, and the parameters $(\boldsymbol{\mu}, \boldsymbol{B})$, which control the opinion updates. As we feed more messages into the estimation procedure, it becomes more accurate.

(a) Temporal parameters, $(\boldsymbol{\alpha}, \boldsymbol{A})$    (b) Opinion parameters, $(\boldsymbol{\mu}, \boldsymbol{B})$

Figure 5: Performance of model estimation for several $512$-node kronecker networks in terms of mean squared error between estimated and true parameters. As we feed more messages into the estimation procedure, the estimation becomes more accurate.

# K Twitter dataset description

We used the Twitter search API[8] to collect all the tweets (corresponding to a 2-3 weeks period around the event date) that contain hashtags related to the following events/topics:

- **Politics:** Delhi Assembly election, from 9th to 15th of December 2013.
- **Movie:** Release of "*Avengers: Age of Ultron*" movie, from April 28th to May 5th, 2015.
- **Fight:** Professional boxing match between the eight-division world champion Manny Pacquiao and the undefeated five-division world champion Floyd Mayweather Jr., on May 2, 2015.
- **Bollywood:** Verdict that declared guilty to Salman Khan (a popular Bollywood movie star) for causing death of a person by rash and negligible driving, from May 5th to 16th, 2015.
- **US:** Presidential election in the United-States, from April 7th to 13th, 2016.

We then built the follower-followee network for the users that posted the collected tweets using the Twitter rest API[9]. Finally, we filtered out users that posted less than 200 tweets during the account lifetime, follow less than 100 users, or have less than 50 followers.

| Dataset | $|\mathcal{V}|$ | $|\mathcal{E}|$ | $|\mathcal{H}(T)|$ | $\mathbb{E}[\mathbf{m}]$ | $\mathbf{std}[m]$ |
|---|---|---|---|---|---|
| Tw: Politics | 548 | 5271 | 20026 | 0.0169 | 0.1780 |
| Tw: Movie | 567 | 4886 | 14016 | 0.5969 | 0.1358 |
| Tw: Fight | 848 | 10118 | 21526 | -0.0123 | 0.2577 |
| Tw: Bollywood | 1031 | 34952 | 46845 | 0.5101 | 0.2310 |
| Tw: US | 533 | 20067 | 18704 | -0.0186 | 0.7135 |

Table 1: Real datasets statistics

# L Baselines description

- **Collab-Filter:** We adopt the proposal described by Kim *et al.*in [19] to use sentiment prediction using collaborative filtering. Let us assume that $U$ and $V$ are two latent low rank matrices capturing the characteristics of *spread* and *reception* of the users. $R_{i,j}^{(t)}$ is the sentiment value posted by user $i$ at time $t$ that evokes a message of user $j$ before any other neighbor of $j$ posts a message. To compute $U$ and $V$, we minimize the following optimization problem.

$$\sum_{t \in \mathcal{H}_t} \sum_{i,j \in \mathcal{V}} I_{i,j}(R_{i,j}^{(t)} - U_i^T V_j)^2 + \lambda_1 \sum_i ||U_i||_2^2 + \lambda_2 \sum_j ||V_j||_2^2$$

  Here $I_{i,j}$'s are the entries of adjacency matrix.
- **Flocking:** In the flocking model [15], a node $i$ with opinion $x$ first selects the set of neighbors $j$ having opinions $x_j$ so that $|x_i - x_j| < \epsilon$ and then updates her own opinion by averaging them.
- **Voter:** In this strategy [26], at each step, a node is selected at random; she chooses one of its neighbors uniformly at random (including itself) and adopts that opinion as its own.
- **DeGroot:** This model [9] suggests that a node updates her opinion by taking a weighted average of her neighbors' opinion. In particular, this proposal assumes that the array of weights form a row-stochastic influence matrix with $w_{i,j} \geq 0$, and opinions in the range $[0, 1]$ (which stochastic updates preserved).
- **Linear:** A generalize linear model [8] generalizes the DeGroot model by relaxing the structure of influence matrix. Under this model, the influence matrix need not be stochastic. Moreover, the influence parameters can take negative values.
- **Biased Voter:** A biased voter model [7] captures various aspects of other opinion models by a mix of three factors: stubbornness (ignoring others opinions), DeGroots weighted averaging with neighbors, and biased conformance (Flocking Model), which selects users having opinion close to that of the base agent.

## Footnotes

[8]https://dev.twitter.com/rest/public/search

[9]https://dev.twitter.com/rest/public