[Reviews · NeurIPS 2016]

Reviewer 1

Summary

In a nutshell, the paper proposes a new model for opinion-dynamics in social networks (where users can publish messages/posts/tweets). The authors discuss how their model can be efficiently trained, used to predict opinion dynamics. The experiments demonstrate that the model has more predictive power compared to existing alternatives. In my view, the following aspects of the model make it stand out among the classical models for opinion dynamics: 1) The opinions of individuals are updated asynchronously. That is, only the opinion of those who observe a new post/message gets updated. 2) The model captures the intentisity of the posted messages which highly impacts how the opinions will develop. On the other hand, most classical models assume that opinions get updated at a constant rate. 3) The model assumes that the observed signals (posted messages) are simply noisy reflections of the underlying opinions. This makes the model more robust and applicable when dealing with real-world datasets. The technical aspects of the paper is mostly concerned with how the proposed model can be viewed as Markov process and how it can be expressed using SDEs. Furthermore, the paper discusses how the model can be trained using existing methods and how it can be simulated. A simulation, in this case, is essentially equivalent to sampling from the proposed generative model which is not trivial, but the authors point out how it can be done using existing algorithms. Finally, the experiments highlight that (a) the model is complex to produce exteremly different outcomes (i.e, consensus vs. polarization), and (b) the model has a higher predictive power compared to existing opinion-dyanmic models. The real-world experiments are performed on datasets obtained from twitter.

Qualitative Assessment

The technical aspects of the paper (i.e., the genarative model, how it can be learned, and how it can be simulated) are quite interesting. The proposed model is a complex data-oriented model (compared to its alternatives) which I expected to be too difficult to study analytically. Nevertheless, the paper talks about both simulation-based and analytical forcasting using the proposed model. As I mentioned earlier, the proposed model has some unique properties that makes it interesting compared to classical models for opinion-dynamics. While I really enjoyed the technical contributions of the paper, I feel the experiments don't do justice to the paper. For instance, 1) the predictive power of the model is only tested on twitter. 2) two (decay) parameters of the model are determined through cross-validation. Is this search easy? What is the appropriate range and granularity for these parameters? Is there a smart way to search this space? 3) It is not really clear how much data is used for training. This is only reported in percentages, but I'm curious to know how many tweets were available when an opinion is forecasted (say in Figure 4). Overall, I think the paper is well-written with significant technical contributions. To me, the experiments can be enriched to further improve the quality of the paper.

Confidence in this Review

2-Confident (read it all; understood it all reasonably well)


Reviewer 2

Summary

The paper studies a model of opinion dynamics in social network. The model is a mix of a Hawkes process that determines the timing of message posts, and a model based on latent variables that determines the sentiment of the messages. It's shown how the parameters of the model can be estimated by maximizing log likelihood and the model is evaluated experimentally with data scraped from twitter.

Qualitative Assessment

The proposed model seems like a reasonable model of opinion dynamics in a social network. However, it is a composition of ideas (Hawkes processes, latent variable model) that are not particularly novel. The problem itself is also not new. Given these, I'd rank the paper below average in terms of its novelty.

Confidence in this Review

2-Confident (read it all; understood it all reasonably well)


Reviewer 3

Summary

The article entitled "Learning and Forecasting Opinion Dynamics in Social Networks" studies the problem of opinion forecasting. It proposes two methods of forecasting, analytical and empirical. The methods are shown to be effective in a set of experiments.

Qualitative Assessment

I find the approach of this paper interesting, and the results are solid. Therefore, I recommend its publication in NIPS.

Confidence in this Review

2-Confident (read it all; understood it all reasonably well)


Reviewer 4

Summary

In this paper, the authors propose a novel opinion evolution model, SLANT, which is a continuous time probabilistic model. It distinguishes two kinds of opinions: latent opinion and sentiment (or expressed opinion). People's latent opinion is fixed and hidden, while their sentiments are shaped by both their latent opinion and their friends' sentiments and can be observed at some time points. The model is general and can fits different forms of intensity for messages and sentiment distribution. The authors propose a method to estimate the model parameters and use this model to do the opinion forecasting. Experiment on Twitter data shows that the model fits well to the date. The model estimation, simulation and forecasting can deal with large graphs.

Qualitative Assessment

Overall, I like the model setting. The authors also give the method of estimation, simulation and forecasting based on the model. The analysis is solid and the experiment is sufficient and scalable. Minor questions: For the model part (Eq. 4), besides the factors you have introduced (i.e., people tend to adopt their latent opinions or their friends' sentiments), have you considered to include the tendency of maintaining their current opinions? Can it be regarded as a kind of combination of the factors you have introduced? In addition, Equation (3) confuses me a little bit, i.e., the second equality in Equation (3). The first confusion is that dN_v(t) including in the right hand of the equation should be a random variable (thus we can take expectation of it like in Equation (2)). However, the left hand of Equation (3) is lambda^*_u(t), which is a normal value or variable, not a random variable. It is meaningless that a normal value equals to a random variable. The second confusion is that k(t)*dN_v(t) should be infinitesimal, right? More specifically, according to the definition of convolution, k(t)*dN_v(t) = \int_{-\infty}^{\infty} k(t-s)dN_v(s)ds. Why is it equal to \sum_ei k(t-t_i)?

Confidence in this Review

2-Confident (read it all; understood it all reasonably well)


Reviewer 5

Summary

This paper presents a model of opinion dynamics. The model represents opinions as latent variables that evolve dynamically according to a stochastic process, and are expressed in messages as measured sentiment values. The model also allows opinions of agents to influence each other through the agents' messages. After describing the model and performing some synthetic experiments, the authors use this model to predict opinion dynamics in Twitter data.

Qualitative Assessment

The problem of modeling and predicting opinion dynamics is an important one, and an underrepresented area that is potentially of broad interest at NIPS. The authors have done a thorough job analyzing and evaluating their model. While Hawkes processes generally have become popular in recent years for representing interaction processes, there is lots of work to be done still. I know of no prior work extending this class of models to opinion dynamics, so the model presented seems reasonably novel. However the novelty is somewhat limited by the fact that the main property that makes this model one of opinion dynamics is that the input data consists of sentiment scores, rather than e.g. unsigned interaction events. Since there isn’t much in the structure of the model that makes it specifically a model of opinion dynamics, other recent work in machine learning using Hawkes processes should be cited. (e.g., Blundell, Heller, and Beck "Modelling Reciprocating Relationships with Hawkes Processes") The authors compare to a number of previous models of opinion dynamics, but it’s not clear these are the most relevant baselines. These prior models were built by and large as scientific models, not as predictive models. Since the proposed model (SLANT) doesn’t seem to add much scientific value (what are the unique insights that can be drawn about opinion dynamics because of this model?), the main proposed added value appears to be predictive power. To that end, the most appropriate baselines seem like they would be other unstructured or weakly structured predictive models, such as Gaussian processes or recurrent neural networks. An additional issue with the experiments on real data is that no details were presented on how the baselines implemented / applied to the data set. This information is critical to evaluating the experimental results, and each baseline should be described at least briefly. (To make space for these descriptions, the authors easily could have moved their propositions and theorems, which added little to the paper, to the appendices.) The authors should also provide more details about their methods. Was the test set completely held out, or how many model iterations were performed on it? And the authors should communicate an understanding of why their model performs better than the baselines. The most impressive result is the Hawkes process in figure 4 a and c. The model there seems to capture the cyclic behavior of the data nicely. How does the model capture that? Is it predicting that totally out of sample, or conditioned on the recent data? Why do the other models fail to capture that pattern? Overall paper seems reasonably solid though perhaps a bit incremental and includes an acceptable number of minor flaws.

Confidence in this Review

2-Confident (read it all; understood it all reasonably well)